# Severe Immune-Related Adverse Events: A Case Series of Patients Needing Hospital Admission in a Spanish Oncology Referral Center and Review of the Literature

**DOI:** 10.3390/diagnostics12092116

**Published:** 2022-08-31

**Authors:** Elia Seguí, Carles Zamora-Martínez, Tanny Daniela Barreto, Joan Padrosa, Margarita Viladot, Javier Marco-Hernández

**Affiliations:** 1Medical Oncology Department, Hospital Clínic de Barcelona, 08036 Barcelona, Spain; 2Translational Genomics and Targeted Therapies in Solid Tumors, August Pi I Sunyer Biomedical Research Institute (IDIBAPS), 08036 Barcelona, Spain; 3Medical Intensive Care Unit, Internal Medicine Department, Hospital Clínic de Barcelona, 08036 Barcelona, Spain

**Keywords:** immune checkpoint blockade, immune-related adverse events, colitis, pneumonitis, myositis, myocarditis, nephritis, hypophysitis, hepatitis

## Abstract

Immune checkpoint inhibitors (ICI) have revolutionized the landscape of cancer treatment. Although several studies have shown that ICIs have a better safety profile than chemotherapy, some patients develop immune-related adverse events (irAEs), which require specialized and multidisciplinary management. Since ICI indications are rapidly increasing, it is crucial that clinicians involved in cancer care learn to identify irAEs and manage them properly. Here, we report a case series of 23 patients with severe irAEs requiring hospitalization over a period of 12 months and seize the opportunity to review and update different general features related to irAEs along with the management of the most frequent severe irAEs in our series.

## 1. Introduction

The introduction of immune checkpoint inhibitors (ICI), such as anti-PD-1/PD-L1 and anti-CTLA-4, has become a new game changer in the treatment of an ever-growing number of cancer types. Anti-PD-1/PD-L1 antibodies alone in monotherapy or combined with anti-CTLA-4 antibodies or different chemotherapy agents have demonstrated unprecedented clinical efficacy and durable responses in more than 15 cancer types in the advanced setting [1,2]. Moreover, clinical trials with ICIs are further expanding to cancer types historically considered immunological quiescent [3,4] and to earlier settings of the disease [5].

Given their different mechanisms of action compared to cytotoxic chemotherapy and targeted therapies, ICIs’ side effects also vary substantially. In fact, due to excessive immunity against healthy organs, the use of these drugs is associated with a wide spectrum of unique immune-related adverse events (irAEs). Any organ can be hypothetically involved, although some irAEs are much more common than others, such as those affecting the skin, endocrine organs, the gastrointestinal tract, the liver, and lungs. Others, such as neurological disorders and myocarditis, are much less frequent, although they can be very severe, even lethal. The exact pathophysiology underlying irAEs is not well known, but recent studies show that T-cell activation, autoantibody production, and cytokine responses might be involved.

Currently, irAEs are managed according to broadly used but not evidenced-based algorithms. Corticosteroids are used in most moderate and severe cases. For steroid-refractory irAEs, though, there is no standard treatment defined, and description in literature is scarce. Since indications for ICI are rapidly expanding, proper training of clinicians in the early identification and prompt management of irAEs is key for the amelioration of immunotherapy side effects. Herein, we report a case series of severe irAEs requiring hospitalization over a period of 12 months in a tertiary referral hospital. Given the continuous advances, the present case series provides the opportunity to review and update several general aspects related to irAEs as well as the clinical management of severe specific-organ-based toxicity. To be noted, only severe toxicities with a frequency higher than 5% in our case series will be reviewed.

## 2. Hospitalization Due to Immune-Related Adverse Events

In 2021, 23 patients with solid organ malignancies undergoing oncologic treatment with immunotherapy (atezolizumab, avelumab, durvalumab, ipilimumab, nivolumab, pembrolizumab, or combination treatment including ICI) needed hospital admission due to irAE in our hospital, which is a Spanish referral center in oncology. In 2021, 3300 patients underwent any kind of cancer treatment in our center (excluding best supportive care), with 361 of those treated with ICI. Thus, around 6% of patients treated with ICI needed hospitalization due to irAE, most of them severe.

When focusing on the underlying oncologic illness, patients suffering from metastatic malignant melanoma or lung cancer were the most frequently affected by severe irAE, which is not surprising since ICPIs are the cornerstone in the standard of care for those malignancies. Pembrolizumab, nivolumab in monotherapy, and the combination of ipilimumab and nivolumab were the most frequently responsible drugs for severe irAE. It is well known that ICI combinations are potentially more toxic than ICPI monotherapy, and nivolumab and pembrolizumab were the most frequently prescribed immunotherapy drugs in our hospital in 2021 (150 and 128, respectively). Table 1 reflects the underlying oncologic disease and the responsible treatment for irAE of our 23 patients needing hospital admission.

Regarding organ toxicity, some patients suffered from more than one irAE at the same time. Specifically, one patient presented with thyroiditis, hypophysitis, and hepatitis, and another one with hypophysitis and colitis. All four patients who suffered from severe myositis had myocardial involvement, but it can be considered part of the same irAE (striated muscle damage), as will be detailed later. The other 17 patients underwent a single organ irAE. Table 2 reflects the damaged organs in the 23 patients.

Patient demographic and cancer data, the treatment used for irAE management, irAE evolution, and cancer outcomes for the 23 patient case series are shown in Table 3.

## 3. General Aspects of Immune-Related Adverse Events

### 3.1. Pathophysiology

The exact pathophysiology underlying immune-related adverse events remains unknown but is believed to be related to the role that both CTLA-4 and PD-1/PD-L1 pathways play in immune homeostasis and the prevention of autoimmune diseases. Research findings indicate that CTLA-4 and PD-1/PD-L1 act in different stages of T-cell activation: while CTLA-4 attenuates T-cell response at a proximal step [6], PD-1/PD-L1 inhibits T-cells at further stages of the immune response and in peripheral tissues [7,8]. Thus, irAEs will differ in patients treated with anti-CTLA-4 from those treated with anti-PD-1/PD-L1, with the effects of anti-CTLA-4 generally being more severe [9].

In most cases, irAEs are thought to be related to autoreactive T cells that bind to shared antigens in both tumor and irAE tissue. In a report of two melanoma patients who died from fatal myocarditis, shared T-cell clones’ infiltration was found in both tumor and heart, with no B cells or antibody deposits identified [10].

However, humoral immunity (B cells and autoantibodies) may also play an important role in certain irAEs, with autoantibodies found in patients with thyroid abnormalities, patients who develop type 1 diabetes mellitus, and patients with induced bullous pemphigoid, among others [11,12,13]. Of note, autoantibody frequency is significantly lower than in patients with the same autoimmune disease who did not receive ICI.

In addition, it is also likely that some irAEs might be caused by enhanced complement-mediated inflammation due to the direct binding of ICIs on normal tissue. For instance, it is known that CTLA-4 is strongly expressed in normal pituitary cells, which may explain the higher incidence of hypophysitis seen with anti-CTLA-4 treatments [14].

Finally, some studies suggest that cytokines and chemokines might also be involved in the pathophysiology of irAEs, with elevated levels of IL-17 found in both patients with ipilimumab-induced colitis and preclinical models of colitis [15,16].

### 3.2. Risk Factors and Predictive Biomarkers

The reason why only certain patients develop irAEs while others do not ever experience them after months of treatment is still not well known. Multiple studies have reported different potential personal risk factors, such as a history of autoimmune disease, high body mass index, and significant kidney disease, among others [17].

Since germline genetic factors are known to be related to some autoimmune diseases, some studies are investigating whether such factors (e.g., HLA genotypes) are also related to the likelihood of experiencing an irAE among patients treated with ICI [18].

In addition, since emerging evidence suggests that the composition of the intestinal microbiota could be associated with immune checkpoint blockade efficacy [19,20], some studies are also investigating whether these variations in the gastrointestinal flora might also influence the risk of developing an irAE [19,20]. In two retrospective studies, a higher relative abundance of the *Bacteroidetes phylum* was shown to be associated with a reduced rate of ipilimumab-induced colitis [20,21]. Further research is warranted to establish if the manipulation of the intestinal microbiota could reduce the risk of colitis and other irAEs.

Recent studies have also explored the role of circulating blood cell counts in predicting the probability of experiencing an irAE. For instance, in a recent retrospective study of advanced NSCLC patients treated with ICI, a low neutrophil-to-lymphocyte (NLR) and platelet-to-lymphocyte (PLR) ratios at baseline were significantly associated with the development of irAEs [22]. Further studies are needed to establish the role of novel predictive biomarkers such as cytokines, microRNAs, and gene expression profiling, among others.

### 3.3. Incidence and Distribution

More than two-thirds of the reported adverse events related to cancer immunotherapy are due to immune-checkpoint blockade [23]. The incidence of irAEs differs depending on the class of ICI used. A recent large meta-analysis showed an all-grade (Grade 1–5) incidence of irAEs in about 83% of patients receiving CTLA-4 inhibitors, 72% of patients receiving PD-1 inhibitors, and 60% of patients receiving PD-L1 inhibitors [24]. In addition, severe irAEs (Grade 3–5) have been reported in 10–27% of patients receiving anti-CTLA-4 and in 7–20% of patients receiving anti-PD-1/anti-PD-L1 [24,25]. Of note, these frequencies increase significantly when ICIs are administered in combination with another ICI (>90% for all-grade irAEs and around 60% for grade ≥3 irAEs) or with chemotherapy.

Furthermore, the irAEs pattern also varies according to the class of ICI administered (PD-1/PD-L1 inhibitors vs. CTLA-4 inhibitors). When compared to PD-1 and PD-L1 inhibitors, CTLA4 inhibitors are more likely to cause colitis, hypophysitis, and dermatitis, while pneumonitis, hypothyroidism, and skeletal symptoms (myalgias, arthralgias) are less frequent [26].

Lastly, irAEs do not seem to be specific to the type of cancer. However, there is some data denoting that patients with different cancer types receiving the same ICI have different frequencies of specific irAEs, which seems to suggest that the differences seen in the tumor immune microenvironment across different cancer types could also induce different irAEs patterns. For instance, when comparing the development of irAEs after anti-PD-1 treatment in patients with melanoma and renal cell carcinoma, a higher frequency of dermatological, skeletal, and gastrointestinal irAEs was observed in patients with melanoma, but there was a lower frequency of pneumonitis [26].

### 3.4. Chronological Patterns

Not only the spectrum of potential target organs affected by irAEs is very broad, but also the timing and temporal evolution. irAEs usually commence within 2 to 16 weeks from the start of treatment but can occur at any time, from only a few days after ICI initiation to even years after treatment completion [27]. Noteworthy, combination therapies are not only associated with a greater risk of irAEs, as described previously, but also with an accelerated onset of irAEs, with a median time to onset of around four weeks [25,28]. For both CTLA-4 and PD-1/PD-L1 inhibitors, dermatologic adverse events are commonly the first to appear, while endocrine irAEs can have a delayed beginning. Pneumonitis and gastrointestinal and liver toxicities, among others, may arise at intermediate points. Even if treatment with ICIs is sometimes given for a long period of time, most studies do not show an increased incidence of irAEs with prolonged treatment. However, later-term toxicity, which will progressively be more relevant since indications are expanding to earlier stages, is still not well known.

### 3.5. Overall Management Approach to irAEs

Prompt diagnosis and intervention are both crucial to avoid worsening to severe or even life-threatening conditions. However, no prospective trials have defined the best treatment approach for effectively managing irAEs. Thus, the clinical practice remains variable and is mostly based on expert consensus guidelines [25,29,30]. Despite not knowing the exact pathophysiology, irAEs arise from excessive immunity toward normal organs. According to the guidelines, glucocorticoids are usually the first-line immunosuppressive agent used to reduce this excessive state of temporary inflammation, and when glucocorticoids are not initially effective, additional immunosuppressive agents can be used. Handling irAEs will often require a multidisciplinary collaboration among oncologists and other medical specialists, who are increasingly becoming aware of these toxicities.

To summarize, for most grade 1 irAEs, ICIs can be continued, and patients often do not require immunosuppressive treatment. On the other hand, grade 2 irAEs typically require temporary withholding of ICIs and close monitoring to decide if systemic steroids need to be initiated (depending on the severity of the target organ affected or if irAEs persist even after withholding ICI treatment). Patients with grade 3–4 irAEs (severe) frequently need to be hospitalized and receive high-dose steroids.

Prednisone is the most frequently used steroid, and its dosing should be adjusted to the severity of the irAE Once started, steroids should be tapered slowly over 4 to 6 weeks. In severe cases, if no improvement is seen after 48 to 72 h or steroids cannot be tapered without a relapse, additional immunosuppressive agents should be considered. A more detailed organ-based toxicity management will be reviewed later.

### 3.6. Impact of irAEs and Immunosuppression on Immune-Checkpoint Blockade Efficacy

Development of an irAE yields evidence of immune system activation following immune checkpoint blockade. Whether this activation is correlated or not with an improved therapeutic response remains somewhat controversial. Even if it is well known that irAEs are not imperative to obtain a benefit from ICIs, increasing evidence suggests that patients who do experience an irAE have better outcomes in terms of response rate, progression-free survival, and overall survival [31]. However, these data are more robust in patients treated with anti-PD-1/anti-PD-L1 inhibitors than those treated with anti-CTLA-4 inhibitors [31,32]. It is also possible that some irAEs are more related to efficacy than others. For instance, multiple studies of melanoma patients treated with immune checkpoint blockade have shown a correlation between vitiligo and better clinical outcomes [33]. However, these data should be interpreted cautiously since most of these studies do not consider the immortal time bias (ITB), which could be crucial since patients who die or have disease progression are less likely to develop an irAE.

Another important issue, since immune checkpoint blockade functions by increasing immunity, is whether the immunosuppression used to treat irAEs may reduce the efficacy of ICIs. Retrospective studies, mainly with melanoma patients, have not reported a loss of efficacy for patients receiving immunosuppression for irAEs [32,34]. However, prospective studies testing immunosuppressive strategies would be needed to answer this question properly. Of note, even if immunosuppression has not been shown to reduce antitumor efficacy, it does increase the risk for other adverse events (e.g., opportunistic infections) that should be weighed [35].

### 3.7. Subsequent Treatments after an irAE: Rechallenging the Immune System

Most irAEs resolve eventually after the initiation of immunosuppressive agents. Thus, one of the main concerns in clinical practice is the safety of restarting ICIs after the resolution of irAEs. Prospective data are scarce since no randomized phase 3 trials have evaluated ICI rechallenge after the resolution of severe irAEs.

Retrospective data have shown that subsequent treatment with PD-1/PD-L1 inhibitors after serious ipilimumab-related AEs is safe and associated with only a 3% of recurrent irAEs [36]. Other retrospective studies [37,38] have shown that between 30 and 50% of patients with a previous irAE during treatment with anti-PD-1/anti-PD-L1 had recurrent or new-onset irAEs when resuming treatment; on the contrary, only 18–20% of patients with a previous irAE during combination treatment (anti-CTLA-4 + anti-PD-1), developed a recurrent or new-onset irAE when resuming treatment with only an anti-PD-1. In these studies, patients with myocarditis or severe neurological irAEs were not included.

Therefore, both the ASCO and ESMO guidelines [25,29] recommend permanent discontinuation for all grade 4 irAEs and for most grade 3 myocarditis, pneumonitis, nephritis, hepatitis, and severe neurological toxicities. For all other patients, the decision to resume treatment should be based on the risk–benefit ratio for each patient, considering the severity of the prior irAE, the possibility of alternative treatments, and the overall clinical context of the patient. It is important to bear in mind that even if it is sometimes safe to resume treatment after an irAE, emerging evidence suggests that many patients will continue to derive benefits from immune checkpoint blockade after discontinuation [39].

In some cases, the irreversible organ damage and/or decline in performance status following a severe irAE will also affect and limit the subsequent lines of treatment.

## 4. Update of Clinical Management of Severe Specific-Organ-Based Toxicity

The vast majority of irAEs are mild to moderate and can be appropriately managed in an outpatient setting. However, around 20% of the cases are severe or life-threatening, requiring even sometimes admission to an Intensive Care Unit (ICU). The estimated incidence of fatal irAEs is around 1% [40,41,42]. In these severe cases, a multidisciplinary approach is strongly recommended [43]. According to the most frequently diagnosed irAEs in our hospitalization series of cases, we aim to briefly summarize the management of the following toxicities:

### 4.1. Gastrointestinal (GI)

Gastrointestinal IrAEs, along with those involving the skin, are the most reported. In most cases, symptoms are mild and can be managed with symptomatic treatment, with or without discontinuation of ICI therapy. Diarrhea and colitis are the most frequently described complications, although toxicities can occur throughout the GI tract. Lower GI toxicity is more frequently seen in patients treated with ipilimumab, an anti-CTLA-4 antibody [44,45].

For lower GI toxicity, the diagnostic workup and management depend on the grading of the severity, following the Common Terminology Criteria for Adverse Events (CTCAE) using a grade 1 (mild) to grade 5 (death) scale [46]. In grade 1 (mild diarrhea, up to four stools per day, or mild increase in ostomy output over baseline), only a stool exam is recommended to exclude infectious agents, along with symptomatic management with electrolyte replacement, oral rehydration, and antidiarrheal agents. In moderate cases (grade 2; 4–6 stools per day over baseline or moderate increase in ostomy output), the diagnostic workup is the same as that in grade 1, and the treatment involves fluid replacement along with high-dose corticosteroids (oral prednisone/prednisolone at a dose of 1 mg/kg/day or intravenous methylprednisolone at the same dose in case of persistent diarrhea after 3–5 days of treatment). ICI therapy should be discontinued [25,46]. Severe cases (grades 3 and 4) are defined when there are seven or more stools over baseline or a severe increase in ostomy output is present. Other symptoms, such as abdominal pain, rectal bleeding, or mucus in stool, can be present, and complications such as intestinal perforation or megacolon can occur. In these cases, a colonoscopy is indicated to observe macroscopic changes and to take biopsies. If a complication is suspected, an abdominal CT scan should be considered. Grades 3 and 4 require hospital admission and prompt fluid-replacement therapy, as well as corticosteroid treatment (methylprednisolone, 1–2 mg/kg/day). ICI therapy must be permanently discontinued. Refractory cases can be managed with infliximab (5 mg/kg) or vedolizumab (300 mg) [24,46]. In addition, it is of utmost importance to exclude infections, including *C. difficile* and parasites, and cytomegalovirus. Other alternative options for refractory cases have been reported in very small sample sizes, such as tofacitinib and ustekinumab [47,48], as well as fecal microbiota transplantation [49]. Upper gastrointestinal toxicity usually appears with lower tract symptoms, although isolated nausea, vomiting, or enteritis without colonic involvement may be present [50]. Diagnosis might be challenging due to the nonspecific nature of these symptoms and because patients on ICI therapy are often receiving other cancer treatments.

Pancreatic toxicity is uncommon (<2% of patients) and usually consists of a transient asymptomatic increase in amylase and lipase levels [51]. Acute or chronic pancreatitis or chronic endocrine or exocrine pancreatic insufficiency are possible manifestations. Management is based on fluid therapy, corticosteroids in moderate to severe cases, and ruling out other causes.

### 4.2. Pneumonitis

ICI-mediated lung toxicity consists of a focal or diffuse inflammation of lung tissue, which is usually called pneumonitis. It is an uncommon adverse event, estimated below 3% in patients receiving ICI treatments for cancer, and is more frequent with anti-PD-1 than with anti-CTLA-4 antibodies. However, around 1% of the patients can suffer a life-threatening lung injury.

The diagnosis of ICI-mediated pneumonitis is usually challenging since patients with cancer can suffer from different entities that can clinically and radiologically behave similarly, such as lymphangitis or infection. This is especially remarkable in patients suffering from lung cancer, in which ICI treatment is acquiring a fundamental role and pneumonitis is more frequent than in other neoplasms or in patients with underlying lung disease [52,53].

The most frequent clinical manifestations are nonspecific, with dyspnea and non-productive cough being the most frequent ones, occurring in half and a third of patients, respectively. Less-frequent symptoms are fever and chest pain. It is important to note that mild forms of pneumonitis can be paucisymptomatic and consist of a radiological finding when follow-up imaging is performed [25].

Grades 3 and 4 of the Common Terminology Criteria for Adverse Events (CTCAE) are those generally requiring hospital admission. Grade 3 is referred to patients presenting with severe symptoms or needing oxygen therapy, and grade 4 is used when respiratory failure, need for tracheostomy, or intubation occurs [25].

The diagnostic approach usually consists of a chest-X ray, a high-resolution computed tomography, and a bronchoscopy with bronchoalveolar lavage (mainly to rule out infectious diseases or oncologic disease progression). The role of positron emission tomography is still unclear. An appropriate clinical and radiological evaluation are the cornerstones for the diagnosis since lung biopsy is not generally performed despite the ESMO guidelines recommending it for severe cases [20]. Clinicians might be reluctant to indicate a transbronchial or surgical biopsy in fragile patients under respiratory distress. Thus, imaging has a key role, and different radiological patterns for ICI-mediated pneumonitis have been described, with organizing pneumonia and nonspecific interstitial pneumonia being the most common ones. Less-frequent radiological findings are hypersensitivity pneumonitis, acute interstitial pneumonia, sarcoid-like patterns, or acute lung distress injury pattern [54,55].

Concomitant or previous chemotherapy treatment, previous radiotherapy, pre-existing lung disease, non-small cell lung cancer, and tobacco are risk factors for developing ICI lung toxicity. ICI lung toxicity generally occurs within the second or third month of ICI treatment, although late-development cases have been described, even after ICI discontinuation [56].

Regarding treatment, ICI permanent discontinuation is generally advised in severe lung toxicity cases. Most patients respond to corticosteroid treatment, generally consisting of 1–2 mg/kg/day prednisone or methylprednisolone. In severe cases (grade 3 or 4), up to 4 mg/kg/day doses are recommended. If no improvement is observed within the first 48 h, intravenous 5 mg/kg infliximab administration should be considered. Alternative immunosuppressive treatments to infliximab in refractory cases are mycophenolate or cyclophosphamide intravenous pulses [57]. When the disease responds to corticosteroid treatment, tapering is usually performed in 6–8 weeks [25].

### 4.3. Muscular and Cardiac Toxicity

Muscle toxicity secondary to ICI treatment mainly consists of an inflammatory disease of the muscle cells, which could be referred to as “ICI-induced myositis”. It is a globally uncommon irAE, with an incidence between 0.1% and 1% of the patients treated with immunotherapy drugs after excluding unspecified myalgia without histologic damage demonstration or unequivocal relation to ICI. Most of the cases occur after the second or third immunotherapy cycle administration, but some cases have been described after being under ICI treatment for months. Despite the low incidence of muscle toxicity, it is important to use an accurate diagnosis algorithm and treatment since ICI-induced myositis can lead to life-threatening situations and even death. As will be detailed later, cardiac striated muscle can also be impaired [58].

Acute or subacute weakness is the main symptom of ICI-induced myositis, affecting the proximal musculature in most patients. However, distal or axial musculature involvement can also be present, even isolated. It is important to perform an appropriate physical examination since it is easy to confuse muscle weakness with asthenia, malaise, or prostration due to the advanced oncologic disease most of those patients are suffering from.

Respiratory muscle involvement is frequent, present in around 40% of cases and representing a life-threatening condition due to the risk of ventilatory failure and hypercapnic respiratory insufficiency. Diplopia, dysarthria, and dysphagia mimicking myasthenia gravis can also occur in one-third of the patients. Myocarditis is not an uncommon complication of ICI-induced myositis. It can present with only electrocardiographic nonspecific abnormalities or troponin elevation, but severe cases can develop systolic dysfunction, arrhythmias, and cardiac insufficiency, which can be life-threatening [59].

Most of the cases described in the literature are severe (grades 3 and 4 of the CTCAE, meaning severe weakness compromising basic activities of daily life and life-threatening conditions, respectively), but this can probably be attributed to publication bias. However, the mortality of ICI-induced myositis is estimated to be around 30% and 50% in diagnosed cases [60,61].

Regarding the diagnosis procedure, a physical examination is the cornerstone to defining the muscle involvement pattern, identifying bulbar impairment, and ruling out other entities that can be easily confused (i.e., myasthenia gravis). Basic blood-test analysis should always be performed, including acute-phase reactants such as C reactive protein and the erythrocyte sedimentation rate, hepatic enzymes, lactate dehydrogenase (LDH), creatine kinase (CK), aldolase, troponin (ultrasensible I troponin if available), and thyroid hormones. Substantial elevations of CK are usually present, with a median value of 7000 U/L, similar to what happens with LDH and aldolase. Rutinary identification of mild or moderate elevation of transaminases in patients receiving ICI therapy can mislead to the suspicion of ICI-induced hepatitis. It is important to remember that aspartate aminotransferase (ASAT) and alanine aminotransferase (ALAT), generally linked to hepatic damage, are enzymes also present in muscle cells. Thus, muscle damage occurring in ICI-induced myositis can also be reflected in blood tests by ASAT and ALAT elevation. Correlation with physical examination and the determination of muscle enzymes is mandatory to avoid mistakes in the diagnostic approach of these patients.

An electrocardiogram to detect the presence of nonspecific alterations, such as repolarization changes or QRS widening, as well as arrhythmias, is mandatory. If dyspnea, tachypnea, or a reduction in arterial oxygen saturation are identified, a determination of arterial blood gasses should be performed. This is recommended in patients with suspected CTCAE grade 3 or 4 myositis. A chest X-ray can also be useful for detecting cardiomegaly, aspiration pneumonia, or atelectasis.

In patients presenting dysphagia, video deglutition or alternative dysphagia tests could be useful to assess their ability to tolerate different kinds of textures in the diet.

Electromyography can be useful to confirm a myopathic impairment pattern and rule out neuropathic entities. It is recommended that the presence of autoimmunity (antinuclear and anticytoplasmic antibodies, complement) is assessed and specific autoantibodies related to autoimmune myositis are determined. The specific autoantibodies are generally not present, but their significance when present is yet unclear. Some hypotheses are that paraneoplastic myositis can also be present in those patients or that underlying autoimmune myositis might be triggered by ICI treatment. Whole-body magnetic resonance imaging is used in some referral centers to evaluate the distribution of muscle impairment and even guide a muscle biopsy.

When myocarditis is suspected, an echocardiogram is mandatory, and cardiac magnetic resonance imaging should be considered if available since it could be more sensitive to detecting subtle inflammatory signs, such as the presence of late enhancement.

A muscle biopsy can confirm the clinical suspicion of ICI-induced myositis, which is usually performed in proximal muscles (deltoid, biceps, or quadriceps). The histologic pattern of this entity is characteristic and combines inflammation, moderate to severe necrosis, and regeneration. The presence of large endomysial, perimysial, and perivascular infiltrates in which aggregate macrophages are present is characteristic, conforming to an almost granulomatous pattern [62].

Treatment is based on immunosuppressive drugs in analogy with autoimmune myositis since specific evidence for ICI-induced myositis is lacking. Transient or, in severe cases, definitive suspension of ICI is recommended. When respiratory muscles are involved or myocarditis is present, life-support treatment in an ICU may be necessary [63].

Corticosteroids are the cornerstone of the treatment, with early introduction recommended in severe cases, even if diagnostic tests have not been yet performed. In patients suffering from CTCAE grade 3 to 4 ICI-induced myositis, a methylprednisolone bolus of at least 250 mg/day for three days followed by 1 mg/kg methylprednisolone or prednisone can be used. The association of intravenous immunoglobulins (IGIV) of 400 mg/day or 5 mg/kg/day for 5 days in severe cases should also be considered.

If an initial favorable evolution is observed, the recurrence risk is lower than in autoimmune myositis with corticosteroid tapering, which can be completed in 4 to 6 weeks. However, the association of a second immunosuppressive drug should be considered, mainly azathioprine, methotrexate, or mycophenolate. When the response is assessed as insufficient by the clinician, the IGIV can be maintained with menstrual periodicity. In refractory cases, rituximab or tocilizumab should be considered as an add-on therapy. Abatacept has been suggested as the treatment of choice when myocarditis is present, and the illness is refractory to corticosteroid and IGIV treatment [64].

### 4.4. Nephritis

Acute kidney injury (AKI) related to ICI therapy has an incidence of around 2–5% [65,66]. Although acute interstitial nephritis (AIN) is the most common type, other renal complications have been reported in large series [67], such as acute tubular necrosis or glomerular disease. As in other toxicities, the combination of drugs, including ipilimumab, increases the risk for ICI-related AKI.

The initial diagnostic approach includes ruling out other causes of AKI, such as volume depletion, contrast-enhanced nephropathy, and obstructive uropathy. Thus, all suspected cases should undergo a complete blood test, a urinalysis, and a reno-vesical ultrasound. Although of intense debate, both the ASCO and the NCCN guidelines recommend immunosuppression therapy initiation without a kidney biopsy unless a moderate/severe renal iRAE defined as Common Terminology Criteria for Adverse Events Grade 2–3 or higher is present [29,68]. However, if significant proteinuria or active urinary sediment is present, performing a kidney biopsy is strongly recommended to rule out other causes of AKI or less-frequent forms of ICI-related AKI.

In all patients with ICI-related AKI suspicion, ICI should be temporarily discontinued until the cause is clarified. If confirmed, those with stages 2 or 3 AKI should initiate corticosteroid therapy if no contraindication is present. The usual dosage is prednisone 0.5–1 mg/kg/day (or equivalent) for stage 2, and intravenous pulses for 3 days of methylprednisolone 1–2 mg/kg, followed by oral prednisone treatment for stage 3. A slow tapering during 8 to 12 weeks is usually required. If corticosteroid refractory or significant side effects are observed, escalation with or switching to other immunosuppressant agents (mycophenolate, infliximab, or rituximab) should be considered.

### 4.5. Endocrine Disorders

Endocrine immune-related adverse events are among the most common ICI toxicities. The most common organs affected in descending order are the thyroid (usually hypothyroid, which may be preceded by transient thyroiditis-induced thyrotoxicosis), pituitary (panhypopituitarism or hypophysitis), adrenal (primary insufficiency), and beta cells of the pancreatic islets (insulin-deficient diabetes). The time of onset and the severity of these toxicities is widely divergent. As manifestations are usually nonspecific, a high index of suspicion is required [69].

Thyroid toxicity is the most frequent, as it occurs in up to 20% of patients on ICI therapy [70]. Hypothyroidism is the main form of presentation, usually preceded by destructive thyroiditis. Patients with pre-existing anti-thyroid antibodies are at higher risk. Thyroid-stimulating hormone (TSH) is the preferred and more sensitive test for diagnosis. Thyroid hormone replacement should be started when TSH is markedly elevated (>10 uIU/mL), and follow-up every 8 weeks or sooner is recommended.

Hypophysitis is a very rare condition outside the context of ICI, but it occurs in up to 10% of patients on this therapy. Clinical presentation is usually related to neuro-compression (headache, nausea, diplopia) and, more often, to secondary adrenal insufficiency. Severe cases can present with hypotension or adrenal crisis. Secondary hypothyroidism and hypogonadism are also quite common. Diagnosis consists of low morning cortisol and ACTH, with low TSH and low free T4, as well as sex hormones (FSH and LH). A brain MRI with pituitary windows is strongly recommended when available, although normal brain imaging does not rule out this entity. Hormone replacement therapy should be indicated, including glucocorticoids, thyroid, and, when indicated, testosterone and estrogen [29].

Primary adrenal insufficiency is rare but can be life-threatening. Laboratory tests reveal decreased cortisol and elevated morning ACTH levels.

### 4.6. Hepatitis

Hepatic toxicity secondary to ICI treatment occurs in around 5 to 10% of patients, being more frequent with anti-CTLA 4 or ICI combinations than anti-PD1 alone [71], with severe hepatitis making 3% of the total. According to CTCAE grading, grade 3 is when transaminases (alanine aminotransferase, ALT; aspartate aminotransferase, AST) or alkaline phosphatase (ALP) are 5–20 times over the upper normal value or total bilirubin 3–10 is times over the upper normal value. Grade 4 would be those in which AST, ALP, or ALP are >20 times or total bilirubin >10 times over the upper normal value. The fact that prothrombin time, INR, albumin, or the development of hepatic insufficiency events such as encephalopathy are not included in the CTCAE criteria could merit discussion, but this is not the purpose of the present review. We suggest taking those conditions into account when evaluating and managing liver toxicity.

ICI-mediated hepatitis usually occurs within the two first six to twelve weeks of therapy, but it can also appear after 6 months since the beginning of treatment [72]. Most patients have no symptoms when hepatitis is suspected, with the detection of abnormal liver tests during follow-up being a red flag for clinicians. Fever, abdominal pain, jaundice, or itching can be present and are symptoms that would suggest hepatic function has to be assessed. The development of hepatic insufficiency symptoms is uncommon [73].

When the diagnostic approach is performed, it is mandatory to think about and rule out other causes of alteration in blood liver tests such as muscular injury, alcoholic hepatitis, the increasingly frequent non-alcoholic fatty liver disease, drug-induced liver damage of other etiologies (i.e., concomitant chemotherapy, analgesic drugs, parapharmacy or herbal products), vascular hepatic compromise (i.e., ischemic hepatitis, splanchnic, or portal vein thrombosis), viral hepatitis (the most common ones being hepatitis virus A, B, C, and E, keeping in mind the less-frequent cytomegalovirus or Epstein–Barr virus, among others), autoimmune hepatitis, or progression of the oncologic disease with liver involvement. The last item on that list is the most frequent cause of liver blood test abnormalities among patients with cancer.

An accurate anamnesis, complete blood tests including viral serologies and autoimmune hepatitis antibodies, and liver ultrasonography are the main initial complementary explorations. An MRI cholangiography should be considered if cholestasis is predominant, and a CT angiography can be useful if vascular complications are suspected, and the ultrasonography is inconclusive regarding vascular assessment. A liver biopsy is useful in severe hepatitis if the etiology is not yet determined by the previous tests. It can be useful to rule out alternative hepatitis causes, evaluate the degree of inflammation, and identify histologic abnormalities that support ICI-mediated hepatitis. Most liver biopsies show panlobular hepatitis with an inflammatory infiltrate mainly composed of lymphocytes and occasional eosinophils [74].

When severe ICI-mediated hepatitis occurs, discontinuation of immunotherapy and corticosteroid treatment are needed. There is no solid evidence regarding the optimal corticosteroid doses, but 1 to 2 mg/kg/day of prednisone or methylprednisolone is generally used. Some authors recommend lower doses and increase to 1–2 mg/kg/day if no improvement is observed in 48–72 h. Mycophenolate, azathioprine, or tacrolimus should be considered if the response is difficult to achieve or liver blood tests worsen when corticosteroids are tapered, and no other cause is identified. Infliximab has been used in some cases. Delay in treatment initiation is not recommended when clinical or analytic signs of hepatic dysfunction are present [25,75].

## 5. Strengths and Limitations

Several limitations must be taken into consideration in this study. First, this is a single-center retrospective cohort, so its usefulness in other sites must be cautiously analyzed. Secondly, the sample size is small, and it includes a wide variety of patients in terms of primary tumor site and pharmacologic agents used. However, to our knowledge, it is the first case series including 23 patients with severe irAEs requiring hospital admission in a real-life experience. Furthermore, it addresses an ever-growing clinical problem with still very important gaps in both prevention and management.

## 6. Conclusions

Immune checkpoint inhibitors (ICI) have revolutionized the landscape of solid tumor treatment, with indications rapidly expanding. Although their safety profile is better than chemotherapy, some patients will develop irAEs due to excessive immunity against healthy organs. Most irAEs resolve after the use of immunosuppressive agents, which can reduce the excessive state of temporary inflammation. However, some irAEs are related to irreversible organ damage and, in some cases, even fatal outcomes. In our case series, three patients had a fatal outcome following an irAE (two after pneumonitis, one after myositis + myocarditis), and one patient required a colectomy due to severe colitis. Future research is warranted to better elucidate the pathophysiology underlying irAEs to develop more effective and precise treatment strategies for immune-related adverse events.

## Figures and Tables

**Table 1 diagnostics-12-02116-t001:** Underlying oncologic disease and ICI treatment received by the admitted patients due to irAE needing hospital admission.

Underlying Solid Organ Malignancy		Responsible ICI Drug	
Malignant melanoma	12 (52%)	Pembrolizumab	10 (43.5%)
Lung cancer	6 (26%)	Nivolumab	5 (21.7%)
Urothelial cancer	2 (8.7%)	Ipilimumab + Nivolumab	5 (21.7%
Breast cancer	1 (4.3%)	Atezolizumab	2 (8.7%)
Renal carcinoma	1 (4.3%)	Ipilimumab	1 (4.3%)
Colorectal carcinoma	1 (4.3%)		

**Table 2 diagnostics-12-02116-t002:** Damaged organs due to irAE needing hospital admission.

Target Organs Affected by irAEs	
Colitis	5 (22%)
Pneumonitis	4 (17.4%)
Myositis + Myocarditis	4 (17.4%)
Nephritis	3 (13%)
Hypophysitis	2 (8.7%)
Hepatitis	2 (8.7%)
Skin toxicity	1 (4.3%)
Aseptic meningitis	1 (4.3%)
Gastritis	1 (4.3%)
Pancreatitis	1 (4.3%)
Arthritis	1 (4.3%)
Thyroiditis	1 (4.3%)

**Table 3 diagnostics-12-02116-t003:** Patients’ characteristics.

Case	Age	Gender	Cancer Type	ICI	irAE	Initial Steroid Treatment	Add-On Treatment	irAE Outcome	Cancer Status (at Data Revision)	ICI Rechallenge	Follow-Up
1	47	F	Melanoma	Ipilimumab + Nivolumab	Gastritis	Prednisone 60 mg/day (mg/kg/day)	Infliximab	Resolution	Stable disease after ICI discontinuation	No	15 months
2	62	F	Melanoma	Nivolumab	Pneumonitis	Prednisone 30 mg/day (0.5 mg/kg/day)	None	Resolution	Progression of disease; best supportive care	No	14 months
3	49	M	Melanoma	Pembrolizumab	Pancreatitis	Prednisone 70 mg/day (mg/kg/day)	None	Resolution	Sustained complete response	No	13 months
4	59	F	NSCLC	Nivolumab	Nephritis	Prednisone 70 mg/day (mg/kg/day)	None	Resolution	Sustained complete response	No	17 months
5	85	F	Bladder	Atezolizumab	Nephritis	250 mg/day methylprednisolone ×3 days, followed by 30 mg/day (0.5 mg/kg/day)	None	Resolution	Progression disease; best supportive care	No	15 months
6	61	M	Melanoma	Ipilimumab + Nivolumab	Pneumonitis	Prednisone 30 mg/day (0.5 mg/kg/day)	None	Resolution	Progression of disease; death	Yes	12 months
7	55	F	NSCLC	Pembrolizumab	Pneumonitis	Methylprednisolone 120 mg/day (2 mg/kg/day)	None	Fatal	Death during hospital admission (ICU)	NA	NA
8	60	F	TNBC	Pembrolizumab	Myositis + Myocarditis	250 mg/day methylprednisolone ×3 days, followed by prednisone 60 mg/day (mg/kg/day)	IVIG and MMF	Resolution	Progression of disease; death	No	11 months
9	72	M	Melanoma	Pembrolizumab	Nephritis	Prednisone 80 mg/day (mg/kg/day)	None	Resolution	Progression of disease; death	No	1 month
10	77	F	NSCLC	Pembrolizumab	Skin toxicity	Prednisone 30 mg/day (0.5 mg/kg/day)	None	Resolution	Progression of disease; death	No	3 months
11	71	M	RCC	Ipilimumab + Nivolumab	Aseptic meningitis	None *	None *	Resolution	Progression of disease; considering ICI rechallenge	No	14 months
12	55	F	Melanoma	Nivolumab	Colitis	Prednisone 50 mg/day (0.5 mg/kg/day)	None	Resolution	Progression of disease; death	No	8 months
13	66	F	NSLCL	Pembrolizumab	Myositis + Myocarditis	250 mg/day methylprednisolone ×3 days, followed by prednisone 60 mg/day (mg/kg/day)	IVIG and AZA	Resolution	Progression of disease; death nonrelated to cancer	No	6 months
14	79	F	Melanoma	Pembrolizumab	Colitis	Prednisone 60 mg/day (mg/kg/day)	None	Resolution	Progression of disease; death	No	6 months
15	58	F	Melanoma	Ipilimumab	Hepatitis	Prednisone 40 mg/day (0.5 mg/kg/day)	None	Resolution	Progression of disease; alternative treatment	Yes	9 months
16	68	F	Colorectal	Pembrolizumab	Colitis + Hypophysitis	Substitutive hormonal treatment; no corticosteroids	None	Resolution	Stable disease; alternative treatment ongoing	Yes **	10 months
17	79	M	Melanoma	Nivolumab	Myositis + Myocarditis	250 mg/day methylprednisolone ×3 days, 1 g/day methylprednisolone ×3 days, followed by prednisone 70 mg/day (mg/kg/day)	IVIG and TCZ	Fatal	Progression of disease; death related to irAE	No	6 months
18	72	M	NSCLC	Pembrolizumab	Arthritis	Prednisone 30 mg/day (0.5 mg/day)	None	Resolution	Progression of disease; death	No	1 month
19	62	M	Melanoma	Ipilimumab + Nivolumab	Hepatitis + Hypophysitis + Thyroiditis	Substitutive hormonal treatment: prednisone 60 mg/day mg/kg, needing for 1 g/day methylprednisolone ×3 days followed by 2 mg/kg/day increase during follow-up	MMF, TCZ, PE, and IVIG	Refractory	Stable disease after ICI discontinuation	No	7 months
20	68	F	Melanoma	Nivolumab	Myositis + Myocarditis	Methylprednisolone 1 g/day ×3 days followed by 90 mg/day (1 mg/kg/day)	None	Resolution	Partial response after ICI discontinuation	No	6 months
21	73	M	NSCLC	Pembrolizumab	Colitis	Prednisone 60 mg/day (mg/kg/day)	Infliximab	Relapsing	Progression of disease; death	No	9 months
22	67	F	Melanoma	Ipilimumab + Nivolumab	Colitis	Colectomy due to intestinal perforation; methylprednisolone 60 mg/24 h (1 mg/kg/day)	None	Resolution	Progression of disease; alternative treatment	No	6 months
23	68	F	Bladder	Atezolizumab	Pneumonitis	Methylprednisolone 250 mg/day ×3 days followed by methylprednisolone 60 mg/24 h (1 mg/kg/day), needing new increase to methylprednisolone 250 mg/day	Infliximab	Fatal	Death during hospital admission	NA	NA

ICI: immune checkpoint inhibitor; irAE: immune-related adverse event; NSCLC: non-small cell lung cancer; ICU: intensive care unit; TNBC: triple-negative breast cancer; IVIG: intravenous immunoglobulin; MMF: mycophenolate mofetil; RCC: renal cell carcinoma; AZA: azathioprine; TCZ: tocilizumab; PE: plasma exchange; * Patient’s preference and spontaneous improvement; ** Suspension after nephritis related to ICI that did not need hospital admission.

## Data Availability

The authors confirm that the data supporting the findings of this study are available within the article.

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
