# Peer review of "Severe Immune-Related Adverse Events: A Case Series of Patients Needing Hospital Admission in a Spanish Oncology Referral Center and Review of the Literature"

_diagnostics, 2022, doi:10.3390/diagnostics12092116_

Round 1

Reviewer 1 Report

Here, this study report a case series of 23 patients with severe irAEs requiring hospitalization during a period of 12 months. First of all, many reviews and articles have reported on the adverse reactions of immunotherapy, so there is nothing new. There is a similar interesting study to refer to(Preexisting autoimmune disease and immune-related adverse events associated with anti-PD-1 cancer immunotherapy: a national case series from the Canadian Research Group of Rheumatology in Immuno-Oncology). The literature on treatment options for adverse reactions to immunotherapy in the past two years should be added to this study. Prednisone is critical in the management of adverse effects of immunotherapy, and discussion of this section should be increased. How immunotherapy adverse reactions affect subsequent treatment should also be discussed.

Author Response

Dear Reviewer,

We are grateful for your time and consideration of this manuscript and we also very much appreciate your suggestions, which have been very helpful in improving the manuscript. All the comments we received on this study have been taken into account in improving the quality of the article, and we present our reply to each of them separately. 

Point 1: The literature on treatment options for adverse reactions to immunotherapy in the past two years should be added to this study.

We have reviewed again the recent literature of irAEs published in the past two years and have add some new bibliographic citations.

Point 2: Prednisone is critical in the management of adverse effects of immunotherapy, and discussion of this section should be increased.

Some minor changes that can be seen in the control-tracked version have been written following this suggestion. Besides, the prednisone role is also discussed in each organ-based toxicity section.

Point 3: How immunotherapy adverse reactions affect subsequent treatment should also be discussed.

We agree with the reviewer that how irAEs affect subsequent treatments is of crucial importance. When reviewing the general aspects of immune-related adverse events, the subsection 3.6 is dedicated to “subsequent treatments after an irAE”, though it is primarily focused on the safety of rechallenging the immune system after an irAE. Should the reviewer consider something else should be added concerning subsequent treatments, please let us know.  

Thank you very much again for the constructive comments that have certainly contributed to improve this review. We hope that these changes to the manuscript will facilitate the decision to publish the study. We are open to consideration of any further comments on our answers.

Sincerely,

The authors

Reviewer 2 Report

Given the different mechanism of action of immune checkpoint inhibitors (ICIs) from cytotoxic chemotherapy and targeted therapies, side effects of ICIs can vary as well. In fact, ICIs can result in unique and often specific adverse events termed immune-related adverse events (IrAEs), which are believed to be determined by an excessive, uninhibited T-cell-mediated immune response. Although ICIs have been reported to present a favorable safety profile, about 10–15% of IrAEs are grade 3 or 4, leading to treatment interruptions and requiring high- dose corticosteroids; these events may include, among others, hepatitis, colitis, nephritis, pneumonitis, dermatitis, endocrinopathies and several others.

Currently, the effect of IrAEs on survival of patients affected by advanced malignancies is uncertain. A plethora of recent retrospective studies have hypothesized that the development of IrAEs in cancer patients may correlate with durable response and survival benefit, although contradictory reports exist. Despite providing interesting data, several studies investigating this association should be interpreted with caution because of an inappropriate methodology. In particular, only a minority of these reports considered the effect of immortal time bias (ITB), a key element in determining the effective association between clinical outcomes and a time-dependent variable. Of note, ITB represents a key element regarding these kind of studies since patients who die or whose disease progresses earlier are less likely to develop toxicity; in fact, these patients probably have not stayed in the study long enough to develop adverse events, or because they discontinued treatment or died due to progressive disease. Conversely, included patients that stayed in the study for a longer time interval have an increased risk to experience toxicities.

Based on these premises, the study assesses a current, timely topic.

We recommend some changes:
- Major revisions are needed. The main strengths of this paper are that it addresses an interesting and very timely question and provides a clear answer, with some limitations. A severe issue is represented by the sample size and by the usefulness of this study; the analysis is limited to a population from a single instituiton with a very small sample size, and authors should further express this point.
- Second, the study included a widely varied patient population in terms of agents as well as primary tumor site and the total number of patients analyzed was relatively small. Thus, the authors should better highlight the limitations of the current paper.
- The background of the changing scenario of cancer immunotherapy should be better discussed in the introduction section, and some recent papers regarding this topic should be included (PMID: 34431725 ;  
PMID: 34894318; PMID: 32911806).
- Association bewteen IRAEs and clinical outcomes? please talk about this topic.
Major changes are necessary before eventual publication.

Author Response

Dear Reviewer,

We are grateful for your time and consideration of this manuscript and we also very much appreciate your suggestions, which have been very helpful in improving the manuscript. All the comments we received on this study have been taken into account in improving the quality of the article, and we present our reply to each of them separately. 

Point 1: The main strengths of this paper are that it addresses an interesting and very timely question and provides a clear answer, with some limitations. A severe issue is represented by the sample size and by the usefulness of this study; the analysis is limited to a population from a single instituiton with a very small sample size, and authors should further express this point.

Point 2: Second, the study included a widely varied patient population in terms of agents as well as primary tumor site and the total number of patients analyzed was relatively small. Thus, the authors should better highlight the limitations of the current paper.

We agree that the problems raised by the reviewer can be seen as a major concern and should be more extensively discussed. Therefore, we have added a subsection entitled “Strengths and limitations”.

Point 3: The background of the changing scenario of cancer immunotherapy should be better discussed in the introduction section, and some recent papers regarding this topic should be included (PMID: 34431725 ;  PMID: 34894318; PMID: 32911806).

We agree with the reviewer that the changing scenario of immune checkpoint blockade with ever-growing indications should be better discussed. We have reviewed again the recent literature of immune checkpoint blockade published in the past two years and have add some new bibliographic citations. We have also included some of the bibliographic citations suggested by the reviewer.

Point 4: Association between IRAEs and clinical outcomes?

We agree with the reviewer that how irAEs are related to clinical outcomes is of crucial importance. When reviewing the general aspects of immune-related adverse events, the subsection 3.6 is dedicated to “Impact of irAEs and immunosuppression on immune-checkpoint blockade efficacy”, primarily focused on discussing whether the development of irAEs might be correlated with a better response and survival benefit. We have also added some discussion about the immortal time bias (ITB), following the suggestion of the reviewer. Should the reviewer consider something else should be added concerning the correlation between irAEs and clinical outcomes, please let us know. 

Thank you very much again for the constructive comments that have certainly contributed to improve this review. We hope that these changes to the manuscript will facilitate the decision to publish the study. We are open to consideration of any further comments on our answers.

Sincerely,

The authors

Round 2

Reviewer 1 Report

The article meets the requirements for publication after careful revision.

Reviewer 2 Report

The authors addressed all the queries and issues we raised.

We recommend Acceptance.